# Advances in Innovative Surgical Implant Manufacturing for Hernia Repair and Soft Tissue Reconstruction

**DOI:** 10.3390/bioengineering12111182

**Published:** 2025-10-30

**Authors:** Stavros Patsouris, Panagiotis Mallis, Efstathios Michalopoulos, Nikolaos Nikiteas, Nefeli Papadopoulou, Michalis Katsimpoulas

**Affiliations:** 1School of Medicine, National and Kapodistrian University of Athens, Mikras Asias 75, 115 27 Athens, Greece; 2Hellenic Cord Blood Bank, Biomedical Research Foundation Academy of Athens, Soranou tou Efesiou 4, 115 27 Athens, Greece; pmallis@bioacademy.gr (P.M.); smichal@bioacademy.gr (E.M.); 3Hellenic Minimally Invasive and Robotic Surgery (MIRS) Study Group, Medical School, National and Kapodistrian University of Athens, Mikras Asias 75, 115 27 Athens, Greece; nikolaos.nikiteas@gmail.com; 4Independent Researcher, 106 80 Athens, Greece; nefpap92@gmail.com; 5Center for Clinical, Experimental Surgery and Translational Research, Biomedical Research Foundation of the Academy of Athens, 115 27 Athens, Greece; mkatsiboulas@bioacademy.gr

**Keywords:** mesenchymal stromal cells, scaffolds, neoangiogenesis, hernia mesh, regenerative medicine, soft tissue reconstruction

## Abstract

Abdominal wall hernias occur in a high percentage of the general population, making prosthetic hernia repair one of the most common surgical procedures. Despite the significant development of surgical techniques and the improvement of surgical meshes, complications still burden the health of patients, as well as the health system. The successful integration of the mesh plays a crucial role in the minimizing the complications associated with hernia meshes. Regenerative medicine focuses on the development of new treatments and applications to heal tissues and organs in order to restore their function. It uses scaffolds that provide physical support and a suitable environment for accelerating repair and proliferation and promoting the regeneration of damaged tissue. Platelet-rich plasma and stem cells are essential tools in regenerative medicine since they have shown efficacy in multiple fields. The main risk factor negatively affecting the survival of any cell type, including stem cells on a prosthetic material, is ischemia. Without the minimum required supply of oxygen, growth factors, and cytokines, it is impossible for cells to successfully proliferate and differentiate. The addition of PRP to a surgical mesh is hypothesized to increase neoangiogenesis in the area, acting as a reservoir of growth factors that will create the right conditions for the proliferation and differentiation of these cells. The aim of the present review is to record experimental studies that have been published where a scaffold or a hernia mesh is coated with PRP, stem cells, or a combination of them for hernia repair and soft tissue reconstruction.

## 1. Introduction

The abdominal wall comprises a well-organized structure consisting of important components of the human body, including the skin, subcutaneous fat tissues, several muscle layers, preperitoneal fascia, and peritoneum. This complex structure is characterized by remarkable tissue strength responsible for postural assistance and abdominal wall pressure stabilization, also playing the role of a physical barrier for the organs of the abdominal cavity. Abdominal injuries such as trauma, infection, congenital conditions, various complications of abdominal surgeries (e.g., laparoscopic surgeries), and neoplastic disorders could result to hernia development.

Abdominal wall hernia refers to the condition of protrusion of internal tissue or organs which can occur through the weakened musculoaponeurotic wall [1]. Abdominal wall hernias occur in a high percentage of the general population, making hernia reconstruction one of the most common procedures in the field of abdominal surgery [2]. Each year, more than 20 million patients are submitted for hernia surgical repair worldwide [3]. The most common hernia is inguinal, representing the 70% of the cases, followed by the umbilical, epigastric, femoral, incisional, etc. [4].

In medical practice, surgical mesh implantation remains a first-line therapeutic strategy for repairing abdominal hernias [5]. The gold-standard surgical method for abdominal hernia repair is the Lichtenstein open “tension-free” technique. In this approach, the implant is securely attached to the aponeurotic tissue after an incision is performed above the hernia defect [1]. Proper surgical mesh implantation and stabilization onto the aponeurotic tissue using the Lichtenstein technique is associated with lower levels of postoperative pain and better quality of life for patients [1]. The role of the surgical mesh is to accelerate the healing process of the abdominal gap and to restore the mechanical stability of the injured site [1]. Nowadays, laparoscopic approaches, such as the transabdominal preperitoneal (TAPP) and totally extraperitoneal (TEP) methods, have gained significant attention by surgeons due to their minimally invasive nature [6].

A common feature in hernia repair is the proper performance of wound healing, primarily to close the gap and reinforce the abdominal tissue and secondarily to avoid any possible postoperative complications, including bacterial infections, hernia recurrence, and tissue adhesions [4]. The process of wound healing consists of four distinct stages: (a) tissue hemostasis, (b) inflammation, (c) cell proliferation, (d) tissue maturation remodeling [7]. Wound healing is a metabolically demanding process that requires increasing amounts of nutrients and oxygen. In the context of wound healing, neoangiogenesis plays a vital role in facilitating the transportation of the necessary nutrients, oxygen, and immune cells [8]. As in every trauma or surgical intervention, a cascade of biological reactions is initiated, where the introduction of a foreign material (surgical mesh) within the patient may trigger an inflammatory immune response [9].

This response is initiated by the initial acute phase reaction, followed by the formation of the implant matrix, primarily facilitated by the migration of fibroblasts responsible for generating sulphated glycosaminoglycans (sGAGs), proteoglycans, and organized collagen bundles. Histologically, the foreign-body reaction (FBR) is characterized by the presence of a granuloma in close proximity to the mesh fiber, surrounded by a collagen capsule that serves as a protective barrier for the host against the foreign material [9]. The extent of this immune reaction during the healing process will determine the degree of integration of the surgical mesh. In this way, a surgical mesh may trigger a severe immune reaction, increasing the chances of postoperative side effects [1]. The development and recurrence of hernias are linked to altered collagen production rates, specifically a reduction in collagen type I and an increase in collagen type III [10]. In this way, modern regenerative medicine and tissue engineering applications may assist in accelerated the wound healing process of abdominal hernia repair, focusing on the combined use of stem cells and growth factors contained in plasma derivatives to manufacture surgical mesh with advanced properties [11].

In the field of tissue engineering, a scaffold with specific structural–functional properties, e.g., pore size, will serve as a proper microenvironment for stem cells, thus promoting cell proliferation and differentiation [12]. The scaffold provides physical support and a suitable environment, accelerating repair and promoting the regeneration of damaged tissue [12]. The ideal scaffold needs to fulfill specific criteria, such as mimicking the natural extracellular matrix (ECM) of the target tissue, enabling cell attachment, migration, proliferation, and differentiation, fostering vascularization and the delivery of nutrients, and being capable of degradation to facilitate the development of new tissue [4,13]. In the case of hernia repair, the surgical mesh is playing the role of the scaffold, promoting the wound healing process [4]. Moreover, the production of advanced tissue-engineered products relies on the proper combination of stem cells with the surgical mesh. The produced surgical mesh with advanced properties will assist to proper hernia repair, eliciting only a mild host immune response against the implanted material [11].

In this review, we will shed light on abdominal wall hernia restoration and soft tissue reinforcement utilizing state-of-the-art applications in regenerative medicine and tissue engineering to develop advanced surgical implants in combination with either stem cells or growth factors originating from plasma-derived products.

## 2. Surgical Mesh, Stem Cells, and Plasma-Derived Products

### 2.1. Surgical Mesh Implants

Surgical mesh implants are typically described as thin, flexible, flat sheets designed to strengthen and provide stability to weakened tissue [13]. Meshes are currently used in a great number of applications in reconstructive surgery including the treatment of hernia, pelvic organ prolapse, urinary incontinence, and others [1]. Starting in 1878, Theodor Billroth proposed mesh implantation as the ideal solution for surgical hernia repair [3]; nowadays, surgical meshes can be manufactured utilizing absorbable, non-absorbable, and biological materials [14].

The majority of the non-absorbable surgical meshes are made from materials such as polypropylene (PP), polyester, expanded polytetrafluoroethylene (ePTFE), or polyvinylidene fluoride (PVDF) [13]. Nowadays, the most commonly-used mesh is made from PP [13]. PP mesh implantation for hernia repair has been associated with various adverse effects, including pain, rejection of the mesh as a foreign body, and adhesion to internal organs which can lead to bowel obstruction and fistula formation [13]. Prosthetic meshes, including knitted PP, polytetrafluoroethylene, polytetrafluoroethylene (PTFE) combined with hyaluronic acid, polyester, and polyglycolic acid, are commonly utilized for hernia repair. The use of these non-absorbable materials in implantation has demonstrated efficacy in restoring abdominal wall integrity and reducing hernia recurrence. Nevertheless, patients undergoing hernioplasty with non-absorbable meshes may encounter foreign-body sensation, stiffness, and discomfort. Additionally, non-absorbable meshes have been linked to severe inflammation, resulting in scar formation, chronic pain, viscera adhesions, bowel fistula, and mesh erosion into the bladder. In contrast to natural tissues, vascular ingrowth is restricted in prosthetic grafts, making it challenging to eliminate bacteria once the graft is contaminated. Consequently, it is advisable to refrain from using prosthetic grafts in patients with bacterial contamination and abdominal cavity infections [1,4]. One of the main disadvantages of non-absorbable materials is the induction of adverse foreign-body reactions. Various studies have attempted to address this issue by enhancing the biocompatibility of these materials. Absorbable meshes are designed to gradually break down and be absorbed by the body over time, reducing the presence of foreign material [13].

Biological meshes, on the other hand, consist of decellularized tissue containing a dense network of collagen and beneficial biological factors that promote and enhance the wound healing process [3]. In recent times, various extracellular matrices (ECMs) have been created, using naturally derived materials, to address hernia and abdominal wall defects. These materials facilitate cell ingrowth, are less prone to infection, and are associated with a lower likelihood of eliciting a foreign-body response. However, it is worth noting that naturally derived materials have a tendency to rapidly degrade and present differences in biomechanical properties compared to the natural host tissues, leading to a reduction in both mechanical and tensile strength over time following implantation. This decline in strength raises concerns in clinical applications, especially considering that adequate tensile properties are essential for grafts. Therefore, it is crucial to not only comprehend the biological response to degradable biomaterials but also monitor the mechanical properties of implanted tissues [14,15].

Over the past few decades, advancements in surgical mesh have concentrated on enhancing the biocompatibility of materials and developing mechanisms to facilitate the transmigration and translocation of the host cellular populations to both accelerate the wound healing process and to prevent the possibility of hernia development. Additionally, various surgical meshes coated with cellular populations, growth factors, etc., have undergone clinical assessment with the goal of diminishing host immune reactions, thus enhancing the biocompatibility of the implanted mesh [11,14,16]. Unfortunately, despite the hundreds of surgical meshes that have been proposed and tested, the manufacturing of the ideal mesh requires further exploration by research groups in the field.

### 2.2. Stem Cells

In recent years, a great number of scientists including physicians, surgeons, and researchers from different disciplines have explored the possibility of improving the surgical meshes used for hernia repair. In the context of reconstructive surgery, improvement of surgical mesh manufacturing towards advancing their properties has been reported [1]. Specifically, materials with specific size, shape orientation, pore size, density, mechanical properties, composition (monofilament or twisted), anisotropy, etc., can now be produced and effectively alter the surface and the properties of surgical meshes to improve the wound healing process [17]. To further accelerate the healing process outlined by the applied surgical meshes, combination with stem cells has been proposed to further advance their wound healing properties and soft tissue reinforcement [13]. Indeed, stem cells have been widely studied over the last 20 years for their tissue regenerative abilities and immunomodulatory properties. Stem cells are different types of cells originating from different sources, which retain the ability to effectively differentiate to other cellular populations [18]. At most times, stem cells can be found in niches within the human body, existing in a poised state (G0 or G1 arrest of the cell cycle) and representing a reservoir of undifferentiated cells [19]. Stem cells are characterized by key specific abilities: they can differentiate to other cell types in response to extracellular stimuli, and they may also influence migration to an injured site and the differentiation of neighboring cells (e.g., bone marrow stroma) to other cell types as part of their intrinsic homeostatic mechanism [20]. In this category, a great number of stem cells have been studied, including embryonic stem cells (ESCs), mesenchymal stromal cells (MSCs), hematopoietic stem cells (HSCs), and amniotic epithelial stem cells (APSCs), to treat pathological conditions; therefore, they have been used in a great number of clinical trials [21]. Among the aforementioned stem cells, MSCs represent the most extensively utilized cellular population for enhancing the properties of surgical meshes [13]. MSCs comprise a mesodermal-originated stem cell population, initially observed in bone marrow (BM) cultures and reported by Alexander Fridenstein in 1974 [20,22]. Beyond BM, MSCs have been isolated from other sources, including the placenta, amniotic fluid (AF), umbilical cord (UC), adipose tissue (AT), milk teeth (MT), menstrual blood (MB), etc. [23]. Based on their isolation sources, MSCs can be distinguished into (1) adult-derived (BM, AT, MT, and MB), (2) embryonic-derived (placenta, AF), and (3) fetal-derived (UC) stem cells, which are further characterized by distinct properties, regarding their proliferation/differentiation potential and immunomodulatory properties [24]. Specifically, embryonic- and fetal-derived MSCs have greater proliferation and stemness potential due to longer telomeres and increased telomerase activity, also accompanied by greater immunoregulatory abilities compared to adult-derived MSCs [24]. Another important aspect is that adult-derived MSCs are characterized by a greater number of gene mutations and epigenetic alterations, due to various life and dietary habits and underlying disorders, compared to embryonic- and fetal-derived MSCs [25]. To precisely define an MSC population, the International Society for Cell and Gene Therapy (ISCT) proposed a number of minimum criteria in 2006. These criteria include (1) a spindle-shaped and plastic adherent property of cells, (2) differentiation capacity to mesodermal lineages, including mostly (a) “osteocytes”, (b) “adipocytes”, and (c) “chondrocytes”, and (3) a specific immunophenotype including the positive expression (>95%) of CD73, CD90, and CD105 and negative expression (<3%) of CD34, CD45, CD11b, and HLA-DR [26,27]. It is noteworthy that MSCs are characterized by low expression levels of HLA class II molecules and by the absence of co-stimulatory molecules, e.g., CD40, CD80, and CD86. This in turn may reflect the fact that allogeneic MSCs may avoid immune recognition by host cells, thus allowing consideration of the use of third-party MSCs derived from fetal or embryonic origin as an alternative therapeutic strategy, especially for severely conditioned patients [28]. In the context of the clinical utility of MSCs, a great number of clinical trials have been performed with the utilization of either autologous adult MSCs or third-party (embryonic or fetal) MSCs. Previously conducted clinical trials in different human disorders, including osteoarticular regeneration [29], immune-related disorders [30], GvHD [31], and others, have indicated the safety and tolerability of both autologous and allogeneic MSCs. Regarding wound healing acceleration, MSCs may beneficially act through their differentiation ability to mesodermal cell lineages, including smooth muscle cells and epithelial cells [32,33]. This in turn could lead to proper ECM reconstruction through the production of properly aligned collagen type I and preservation of a high collagen I/III ratio [34,35]. A great body of evidence from the literature has shown that patients with a low collagen I/III ratio have an increased risk for hernia recurrence [36]. When scar tissue is formed, fibroblasts and myofibroblasts contribute to the higher production of collagen type III, resulting in increased strength of the abdominal wall [36]. However, myofibroblasts are diminished by the production of elastin and sulfated glycosaminoglycans (sGAGs), thus decreasing the elastic modulus and biomechanical strain of the formed abdominal wall layer [37,38]. These phenomena may lead to deterioration of the abdominal wall, required in most timed new reconstructive surgery for hernia repair. This is supported by the study of Roy et al., who presented that SMCs are connected with the collagen and elastin sheath, thus providing elasticity to the abdominal wall ECM and enabling it to be resistant to different stress exposures [39]. This fundamental property of SMCs cannot be attributed to the myofibroblasts, resulting in the production of a stiffer ECM, which could be vulnerable to different stress exposures. On the other side, MSCs can properly accelerate the wound healing process by (1) producing several growth factors including transforming growth factor (TGF)-β1, fibroblast growth factor (FGF), platelet-derived growth factor (PDGF), and vascular endothelial growth factor (VEGF) with paracrine action on the resident cells to produce ECM components (including collagen type I, sGAGs, elastin, fibronectin, and perlecan), (2) directly producing ECM components required for the abdominal wall reconstruction, and (3) differentiating to smooth muscle cells and other supporting cells as a response to microenvironment stimulation to enhance tissue reconstruction [40,41]. Beyond the production of growth factors, MSCs possess important immunomodulatory properties which significantly contribute to the hernia repair process [42]. MSCs upon exposure to inflammatory signaling cues, mediated by IFN-γ, TNF-α, IL-1b, and others, can effectively acquire an immunoregulatory phenotype to tolerate immune responses. It has been previously shown that IFN-γ stimulates MSCs through the secretion of prostaglandin E2 (PGE2), IL-1 Receptor agonist (Ra), IL-6, and HGF, leading to a macrophage shift from the M1 to the M2 phenotype. M2 macrophages are characterized by high levels of IL-10 and low levels of IL-12p70, TNF-α, and IL-17 [43]. Gur-Wahnon et al. showed that the cell contact interaction of MSCs with M2 macrophages results in the activation of STAT3, further enhancing the IL-10 mediated immunosuppression of overactivated T cells [44]. The M2 macrophage phenotype also favors tissue remodeling and reconstruction during the wound healing process [43]. Additionally, MSCs can inhibit immune responses through direct cell–cell contact and the utilization of cell death signaling pathways, including the Fas–Fas ligand, PD-PL1, and TNF-TNFR, resulting in the apoptosis of dendritic cells (DCs) and activated T and B cells [24,45]. The inhibition of immune responses is promoted through the secretion of anti-inflammatory cytokines, including IL-1Ra, IL-10, and IL-13, growth factors with known immunosuppressive potential such as TGF-β1, FGF, PDGF, and HGF, and other immunomodulatory biomolecules such as indoleamine 2,3 dioxygenase (IDO), nitric oxide (NO), galectins, PGE2, HLA-G (G1-G7 isoforms), and exosomes containing micro-RNAs (miRs) such as miR-21-5p, miR-142-3p, miR-223-3p, miR-126-3p, miR-145, miR-146, and miR-155 [24,45]. The aforementioned evidence indicates the beneficial use of MSCs in the acceleration of the wound healing process, which can be further combined with surgical implants to advance their properties (Figure 1).

On the other hand, other stem cells such as ESCs and HSCs have exhibited limited use in coated surgical meshes. Specifically regarding the use of ESCs, strong bioethical limitations currently exist, although the possibility of teratoma formation impairs their application in hernia repair and soft tissue reconstruction [46,47]. HSCs, through differentiation to epithelial cells, may exert a potential to strengthen the abdominal wall; however, their application is limited due to the invasive procedure of their isolation to which the patient must be submitted and also due to the requirement of challenging in vitro handling by experienced personnel [48,49]. For all of these reasons, MSCs are the most applied stem cells for surgical mesh coating aimed toward hernia restoration.

### 2.3. Plasma-Derived Products

Lately, platelet derivatives such as platelet-rich plasma (PRP), platelet lysate (PL), leukocyte-rich PRP, leukocyte-low PL, and fibrin gel have been used in a great number of applications, including osteoarticular disorders, burn wounds, diabetic foot ulcers, pediatric epidermolysis bullosa, etc. [50,51,52,53]. These platelet derivatives are characterized by an increased concentration of platelet-derived growth factors, which exhibit significant regenerative action towards the damaged tissue [54]. Recently, a number of studies have shown the anti-inflammatory and regenerative action of platelet derivatives in abdominal wall restoration and hernia repair [55,56,57]. The platelet derivatives may be either autologous or allogeneic in origin; however, in the allogeneic setting a number of necessary tests for infectious disease markers (IDMs) including HIV, HBV, HCV, syphilis, and endotoxin should be performed prior its application to patients [58]. On the other hand, autologous platelet products may have limited use in elderly people or in patients suffering from thrombocytopenia or diseases affecting platelet metabolism, where repetitive blood sampling cannot be performed. The key elements for the regenerative action of platelet products are the growth factors stored in the α-granules of platelets [59]. Therefore, the concentration of platelets in the PRP products should be 3–5-times greater than the initial value determined in the blood sample [60]. Platelet derivatives are rich in a great number of growth factors, including TGF-β1, PDGF-AA/BB, FGF, IGF, VEGF, HGF; cytokines, e.g., TNF-α, IL1α, IL-1β, IL-2, IL-6, IL-8, IL-1R; chemokines such as CCR1; and other biomolecules, e.g., VCAM-1 and ICAM-1 [61]. The combination of those growth factors can accelerate the wound healing process and the attraction and differentiation of cells, as well as potentially promoting immunomodulatory actions in terms of assisting the restoration of the damaged area [62,63]. Platelet derivatives such as PRP and leukocyte-rich PRP can easily be produced through a two-step centrifugation process; however, if the final product is submitted to repetitive freeze–thaw cycles, this eventually leads to platelet lysis and the production of PL [64]. Moreover, if 10% *v*/*v* calcium gluconate, thrombin, or batroxobin is added, followed by incubation at 37 °C for at least 15 min, the formation of fibrin gel will occur [65]. Beyond the autologous obtained derivatives, umbilical cord blood (UCB) has recently been proposed as an equal and alternative source for the production of platelet products [66]. In previously published works, it has been shown that UCB-PRP and PL can maintain the cell culture of MSCs, while it also can lead to their differentiation into vascular SMCs when seeded on decellularized arterial matrices [67,68,69]. Indeed, it has been shown that UCB-PL contains equal levels of growth factors compared to peripheral blood [62,67]. UCB-PRP and PL can be considered as advantageous products for peripheral blood-platelet derivatives, due to the high concentration of HLA-G [70]. Capittini et al. demonstrated that cord blood contains high levels of soluble HLA-G, possibly secreted by MSCs and CD34 + HSCs [71]. HLA-G belongs to the non-classical HLA class I molecules, located at 6p21.3, and is associated with feto-maternal tolerance. HLA-G consists of seven isoforms, the membrane-bound (HLA-G1–G4) and soluble isoforms (HLA-G5–G7) [72]. The mediated immunoregulation is performed through the binding of HLA-G with (a) Immunoglobulin-like transcript 2 (ILT2), located on the cellular membrane of monocytes, NK, T, and B cells; (b) ILT4, found primarily on myeloid DCs and monocytes; and (c) KIR2DL4, found on NK cells [72]. The above interaction can lead either to the apoptosis of overactivated immune cells or to phenotype polarization from effector to immunoregulatory, thus further contributing to halting inflammatory responses [72]. Based on a great body of evidence in studies conducted in both animal models and patients, the implantation of surgical meshes coated with platelet derivatives has clearly demonstrated the safety and beneficial effects, in terms of accelerated wound healing, reduced immune response, and limited hernia recurrence [73,74,75,76,77]. In this way, coated surgical meshes with either autologous or allogeneic platelet derivatives could eventually represent an alternative strategy for the better management of abdominal wall defects, which can be translated into a clinical setting [57] (Figure 2).

## 3. Surgical Mesh with Advanced Properties

### 3.1. In Vivo Studies of Surgical Meshes Combined with Stem Cells

The first study for the application of adipose-derived mesenchymal stem cells (ASCs) on surgical mesh was conducted by Altman et al. [78], using rats subjected to surgically induced ventral hernia repair. ASCs were either pre-seeded onto a porcine acellular dermal matrix or locally injected into the implanted mesh. Two weeks after the implantation process, an augmentation in leukocyte and vascular infiltration in the abdominal defects was noted.

In a study conducted by Lyyanki et al. [79], adipose-derived mesenchymal stem cells (ASCs) were seeded onto a Strattice^®^ mesh for a period of 24 h, before being used to repair induced ventral hernia defects in rat models. The researchers observed that vascular infiltration was more pronounced in the group where ASCs were seeded onto the mesh, four weeks after the surgery.

In a study by Zhang et al. [80], a tissue-engineered mesh (TEM) for repairing inguinal hernias in a rabbit model was produced. This mesh was developed by introducing autologous mesenchymal stem cells (MSCs) into an organized nanofibrous extracellular matrix (ECM). Cells harvested from the bone marrow of New Zealand White rabbits were seeded in a decellularized aorta (DA) scaffold, leading to robust growth and proliferation. The MSCs adhered well to the scaffold, covering it within five days and reaching a high density within the DA at 14 days. In the in vivo part of the study, hernia defects were created alongside the spermatic cord of fifteen adult male rabbits, which were then divided into three groups: the first group underwent hernia repair using TEM, the second group underwent hernia repair using acellular mesh (AM), and the third group received no hernioplasty. After two months, labeled MSCs were found in TEMs, and staining revealed capillary formation suggesting vascularization with some MSCs differentiating into endothelial cells and forming capillaries. TEMs exhibited more significant vascular formations compared to AMs. Moreover, MSCs efficiently prevented adhesion formation. The authors concluded that the new tissue, with a developed blood supply, had a positive impact on the tensile strength and biocompatibility of the scaffold.

In a study by O. Guillaume et al. [81], macroporous meshes were coated with Stromal Vascular Fraction (SVF) cells obtained from human adipose tissue. An onlay model was implemented, which involved creating two muscle defects measuring 0.5 × 0.5 cm^2^ on each side of the linea alba. Subsequently, these hernia defects were repaired using either an SVF-coated mesh or a mesh coated with cell-free fibrin, which served as a control. After 10 or 21 days, the animals were sacrificed. The quantification of vessel volume relative to tissue volume indicated a decrease between day 10 and day 21, but this decrease was observed specifically in the SVF-coated group.

In a study by Federica Marinaro et al. [82], allogeneic bone marrow-derived mesenchymal stem cells (BM-MSCs) were obtained from a Large White pig and used in surgical procedures on 10 pigs that had congenital abdominal non-incarcerated hernias. Monofilament polypropylene (PP) meshes were used to repair the hernias, with some meshes seeded with porcine bone marrow-derived MSCs using a fibrin sealant solution, while others served as controls without MSCs. After one month of mesh implantation, there was a slight but significant reduction in the expression of vascular endothelial growth factor (VEGF) in the group that received MSC-seeded meshes. VEGF is known for its role in promoting angiogenesis, and it is typically associated with wound healing. In summary, this study showed that the presence of MSCs on surgical meshes had a minimal impact on vascularization and led to a reduction in VEGF expression.

Zhao et al. [83], conducted a study in which they seeded autologous bone marrow-derived mesenchymal stem cells (BM-MSCs) from rabbits onto decellularized dermal scaffolds. This approach was used to treat abdominal wall defects in New Zealand white rabbits. They compared animals that received surgical meshes co-administered with MSCs to other animals treated with acellular meshes. The gross examination of the results revealed no recurrence of abdominal hernias and fewer adhesions in the group where MSCs were present. Furthermore, histological analysis indicated better tissue regeneration, increased cellular infiltration, and enhanced angiogenesis in the same group, demonstrating the positive impact of MSCs on the healing process.

In a study by Costa de Oliveira Souz et al. [84], mesenchymal stem cells (MSCs) were isolated from the adipose tissue of rats. Seventy male Wistar R. norvegicus rats weighing 300 g were used for the experiment. The rats were anesthetized, and then their skin was exposed to a heated iron bar to create deep second-degree burns with a diameter of 2 cm. The rats were divided into groups and received membranes coated with or without MSCs that were used as a control. The results indicate that MSCs effectively improved the healing process by promoting angiogenesis and vascularization, modulating the immune response, and inducing epithelialization in the injured area. The use of scaffolding materials combined with MSCs led to functional healing, reduced deformities, effective vascularization, and aesthetically improved outcomes for skin wounds.

Shayanti Mukherjee et al. [85], studied the possible effects of coating a mesh with MSCs for pelvic floor repair. SUSD2+ endometrial mesenchymal stem cells (eMSCs) were extracted from endometrial biopsies collected from seven women who underwent laparoscopic surgery. NSG mice were divided into two experimental groups: one group receiving the nanomesh alone (referred to as “P”) and another group receiving the nanomesh coated with eMSCs (referred to as “P+eMSC”). The procedure involved making a 1.2 cm longitudinal skin incision in the lower abdomen of the mice. The skin was gently separated from the fascia through blunt dissection, creating two pockets on each side of the midline. Nanomeshes were implanted into these pockets in each animal. After either 1 or 6 weeks, the animals were euthanized for analysis. At the 6-week mark, the researchers observed a significantly higher expression of key angiogenic factor genes including Vegfa, Fgf1, Ctgf, Ang1, and Pdgfa in the group that received P+eMSCs, compared to the group that received P alone. The eMSCs were found to reduce acute inflammation, enhance extracellular matrix (ECM) synthesis, promote angiogenesis and neovascularization, and increase the expression of anti-inflammatory genes at the 6-week mark.

A study by Ulrich et al. [86], aimed to analyze the biomechanical characteristics of a mesh coated with gelatin and seeded with eMSCs. A rat wound model was utilized as a preclinical model for pelvic organ prolapse (POP) repair surgery. A total of 74 CBH-run/Arc immunodeficient nude rats were randomly divided into two experimental groups (37 rats/group) and implanted with PA + G meshes, either with eMSCs or without (control). A longitudinal 30 mm skin incision was performed along the spine in the middle of the dorsum. Subcutaneous pockets were achieved by blunt dissection and each animal received either eMSC-seeded or unseeded meshes. Following recovery, the animals were monitored daily until they were sacrificed at 7, 14, 30, 60, and 90 days. Neovascularization occurred more rapidly in the rats treated with eMSCs as shown by the significantly higher number of SMA-positive vessel profiles in the surrounding tissue at day 7. eMSCs were not detectable at day 90 which suggests that these cells influence the healing microenvironment through their paracrine effect, indicating that eMSCs may have prolonged anti-inflammatory effects. In summary, meshes seeded with eMSCs were well-tolerated, encouraged tissue integration, and mitigated long-term inflammatory responses to the implanted mesh. This indicates the possibility of a future treatment option for pelvic organ prolapse (POP).

A study by Xiaolong Zhou et al. [87], aimed to evaluate and enhance the therapeutic effects of adipose-derived stem cells (ASCs) on burn wound healing in a rat model. ASCs were isolated from each rat one month before the experiment and expanded in vitro. A 2 cm^2^ burn wound was made on the dorsal skin of each rat using a heating iron. The rats were divided into different treatment groups. Group 1: Rats received a single injection of 2 × 10^6^ green fluorescent protein (GFP)-labeled autologous ASCs in 500 μL of phosphate-buffered saline (PBS) on day 0, 4 h after wound modeling. Group 2: Rats received three doses of 2 × 10^6^ GFP-labeled ASCs, injected on day 0 (4 h after wound modeling), day 4, and day 8. Group 3 (Control): Rats with third-degree burn wounds received a subcutaneous injections of 500 μL PBS. The rats were euthanized on day 33 for analysis. High levels of vascular endothelial growth factor (VEGF) were found in the wound skin of ASC-treated rats, indicating the role of ASCs in promoting blood vessel formation. Also, it was found that the group treated with adipose-derived mesenchymal stem cells (ASCs) showed significantly better vascularity compared to the group treated with phosphate-buffered saline (PBS). The use of Masson trichrome staining and CD31 immunohistochemistry staining further supported these findings. In summary, the study demonstrated that autologous ASCs, particularly when administered multiple times, had a significant positive impact on angiogenesis and the wound healing process.

A recent metanalysis conducted by Fan et al. [88], showed that cell-seeded surgical meshes strengthened the damaged abdominal wall through the secretion of the aforementioned growth factors, induced collagen production, reduced fibrosis formation, and favored neovasculogenesis, thus resulting in a lower percentage of hernia recurrence events. Moreover, the same metanalysis showed that stem cells effectively tolerate inflammatory responses through the paracrine action of secreted anti-inflammatory cytokines and also promote M2 macrophage polarization. In terms of abdominal organ adhesion, no significant alteration was observed when cell-seeded meshes were applied. However, a number of studies have indicated that MSC-seeded meshes could result in monolayer formation through epithelial cell differentiation, similar to the native peritoneum, in order to prevent the adhesion of the organs.

Research has shown that the transplantation of bone marrow-derived mesenchymal stem cells after a myocardial infarction (MI) can enhance cardiac performance, regenerate the infarcted myocardium, and stimulate angiogenesis in the ischemic area. When administered to cardiomyocytes beating asynchronously, BM-MSCs have the capacity to restore synchronization. In the study of Naan F. Huang et al. [89], human BM-MSCs were cultured and female nude rats underwent left anterior descending (LAD) coronary artery occlusion for 17 min, followed by reperfusion. Five weeks after myocardial infarction (MI), a midline thoracotomy was performed and 50 μL of 0.5% bovine serum albumin for group 1, 2 × 10^6^ cells in 0.5% BSA for group 2, 50 μL of fibrin for group 3, or 2 × 10^6^ cells in fibrin for group 4 was intramuscularly injected into the infarct scar. After 2 days or 5 weeks following the treatment, animals were euthanized and their hearts were immediately frozen for histological and immunohistochemical analysis. The capillary density (total CD31-positive microvessels/mm^2^) and the arteriole density (total SMA-positive microvessels/mm^2^) within the infarct area were evaluated. The quantification of capillaries demonstrated a notably higher capillary density in the BM-MSCs in fibrin group 4 compared to the other groups. A noteworthy increase in capillary density was correlated with elevated levels of VEGF expressed within the infarct scar. The heightened expression of VEGF in BM-MSCs + fibrin (group 4) suggests potential cardioprotective effects on cardiomyocytes and their positive use in regenerative medicine.

The main findings from in vivo studies of surgical meshes combined with stem cells are summarized in Table 1. 

### 3.2. In Vivo Studies of Surgical Meshes Combined with Plasma-Derived Products

Based on the above evidence, mostly PRP and PL can be used for soft tissue reinforcement and hernia repair. The initial results that arose from animal models depicted the beneficial use of coated surgical meshes with platelet derivatives in abdominal wall reconstruction following hernia repair. Indeed, all studies conducted in animal models showed newly formed collagen fibers, microscopic vessel formation, greater tissue thickness, reduced topical inflammation, and limited hernia recurrence events. Following the promising results from animal models, initial application in clinical medicine was then conducted.

Specifically, in the study of Paranyak et al., only 3.7% of the patients undergoing large laparoscopic repair of large hernias with the utilization of PRP-coated meshes exhibited hernia recurrence after 48 months of follow up [76].

In addition, in the study of Popescu et al. [77], 32 patients were enrolled and divided into three groups: (a) golden standard procedure-control group, (b) surgical mesh coated with fibrin gel, (c) surgical mesh coated with PRP. The results showed accelerated wound healing and a mesh integration rate of up to 65% in the groups using platelet derivatives compared to the control group.

Moreover, James et al. [90], launched a study involving 12 patients undergoing large paraesophageal hernia repair utilizing meshes coated with PRP and presented good reflux control accompanied by no postoperative complications or hernia recurrence.

In a study by Fernandez-Moure et al. [91], 48 male Lewis rats weighing around 300–315 g were utilized for the creation and repair of abdominal defects. The rats were randomly divided into two groups: one group received platelet-rich plasma (PRP), and the other group received a saline solution. PRP was extracted from 10 male Lewis rats via blood collection. A 3 cm midline incision was made in the skin and extended through the subcutaneous tissue and a 2 cm incisional defect was made. Subsequently, 3 × 3 cm of porcine acellular dermal matrix (pADM) mesh, with a 5 mm overlap, was used for abdominal wall restoration. In the experimental group, 200 μL of PRP was activated with thrombin and directly administered to the exposed front surface of the mesh. In the control group, 200 µL of saline was applied to the mesh. Closure of the skin incision was achieved using skin staples. The study demonstrated that the application of PRP to surgical meshes led to increased microvessel formation, elevated expression of VEGF, and enhanced angiogenesis, which may contribute to improved tissue repair and healing in abdominal defects.

In a study by Jeffrey L. Van Ep et al. [16], 28 male Lewis rats weighing between 250 and 300 g were divided into two groups to undergo chronic VHR. All rats successfully underwent a hernia-creating procedure. These defects were allowed to mature and become chronic over a minimum incubation period of 28 days. In the control group rats received only the pADM mesh, while the other group received pADM augmented with autologous platelet-rich plasma (PRP). At the 3-month follow-up point, the rats that received PRP augmentation displayed more pronounced visible formation of new blood vessels (neovascularization) within the implanted mesh compared to the rats that did not receive PRP. This visual observation was further confirmed by microscopic analysis during histologic examination. The samples treated with PRP consistently exhibited an enhanced process of neovascularization, signifying improved blood vessel formation within the mesh material when compared to the control group. Furthermore, the PRP-treated samples showed a reduction in the presence of chronic inflammatory cell infiltrates, indicating a less inflammatory response to the mesh and better incorporation of the mesh with native tissue. Overall, the study demonstrated that the addition of autologous PRP to the Strattice™ pADM for chronic ventral hernia repair led to improved neovascularization, reduced chronic inflammation, and a more favorable tissue response at the interface between the native tissue and the mesh implant.

In an experiment conducted by Fernandez-Moure et al. [92], a total of forty-two male Lewis rats, with an average weight of approximately 300–315 g, were subjected to ventral hernia repair using a non-crosslinked porcine acellular dermal matrix. To obtain an adequate amount of blood for the experiment, ten Lewis rats were utilized for the isolation of platelet-rich plasma (PRP). The rats were then randomly assigned to receive either 200 μL of PRP or 200 μL of saline (placebo) during the closure procedure. The surgical procedure involved creating an full-thickness abdominal wall defect measuring 2 cm in length in all rats. The skin was sutured back together, leaving the defect unrepaired and allowing it to develop for 28 days. After this period, the animals underwent hernia repair. A piece of Strattice, with a 5 mm overlap between the muscle and mesh, was used for the repair. The animals were euthanized at 2, 4, and 6 weeks after hernia repair. The results showed that the groups that received PRP exhibited larger, more abundant, and macroscopically visible vessel growth compared to the control group. Extensive vessel ingrowth was consistently observed in the PRP-treated groups at all time points, with the most significant effects seen at 6 weeks. Immunohistochemical staining of CD31, a marker for vascular endothelial cells, showed increased staining in the PRP-treated groups at all time points. In summary, the addition of PRP resulted in significant neovascularization and tissue deposition, particularly when combined with Strattice. The enhanced early neovascularization observed is likely to lead to the development of larger blood vessels at later time points. This suggests that PRP can play a beneficial role in promoting vascularization and tissue healing in ventral hernia repair.

In a pilot study by Cristian E. Boru et al. [93], an iatrogenic hiatal defect up to 4 cm^2^ was made with laparoscopic techniques on 14 female pigs who were divided into two groups. In group A, a Bio-AVR mesh was applied to reinforce the hiatal defect, while in group B the hiatal defect was reinforced using autologous platelet-rich plasma (PRP). After seven months, specimens from both groups were retrieved for histopathological examination. In group B (PRP), the inflammatory response was less significant compared to that in group A. Also, neovascularization of various calibers was noted, along with aspects of active hyperemia suggesting enhanced blood flow. PRP was found to enhance several processes, including angiogenesis, formation of collagen fibers, recruitment of myofibroblasts, and tissue ingrowth. These findings suggest that PRP may have potential applications in promoting tissue repair and healing in such defects.

In a study by Meutia AP et al. [94], a full-thickness skin defect model was utilized to investigate wound healing progress following skin injury. The findings demonstrated that, in comparison to the control group, the PRP-treated group exhibited a notably accelerated rate of wound closure. Furthermore, PRP was observed to have a mitigating effect on wound inflammatory responses. Additionally, PRP was found to promote angiogenesis in the wound tissue. This was evidenced by a significant increase in the amount of new blood vessel formation in the PRP-treated group as compared to the control group. These findings were supported by the assessment of specific markers such as CD31, which is utilized to evaluate vascularization and angiogenesis, as well as vascular endothelial growth factor, a critical growth factor responsible for vascular endothelial cell division and angiogenesis within the wound tissue. 

The main findings from in vivo studies of surgical meshes combined with plasma-derived products are summarized in Table 2. 

### 3.3. In Vivo Studies of Surgical Meshes Combined with Plasma-Derived Products and Stem Cells

The combination of PRP and MSCs has been studied in some cases such as full-thickness wound healing, skin burns, and tendon repair with positive results in the levels of tissue remodeling compared to each component alone (PRP alone, Stem cells alone, or PRP and stem cells).

Yiming Gao et al. [95], conducted a study in which they investigated the enhancement of skin graft texture using adipose-derived stem cells (ADSCs) embedded in platelet-rich plasma (PRP) scaffolds in a rat full-thickness wound model. Briefly, adipose tissue was collected from male Lewis rats and the abdominal aorta of the donor rats was punctured to obtain 8–10 mL of whole blood, which was subsequently centrifuged to obtain PRP. A 2.5 × 2.5 cm^2^ piece of skin in the middle of the back was excised in each rat used in the study. The rats were then divided into four groups: a control group, ADSCs treatment group, PRP gel treatment group, and PRP gel + ADSCs treatment group. For the control group, rats were injected with 1 mL saline under the skin grafts. For the ADSCs treatment group, 10^6^ cells in 0.5 mL saline were injected under the skin graft. For the PRP gel treatment group, PRP gel was applied under the skin graft, and for the PRP gel + ADSCs treatment group, 10^6^ cells embedded in 1 mL PRP gel were applied under the skin graft. Laser perfusion imaging revealed a stronger blood flow signal under the skin graft of the PRP + ADSCs group compared to the control group at two and four weeks after surgery. Moreover, immunohistochemistry results indicated that the number of subcutaneous neovascular (CD31-positive) cells in the PRP + ADSCs group was significantly greater than that in the control group from two weeks after surgery. The RT-PCR results demonstrated a significant increase in the expression of vascular endothelial growth factor (VEGF), basic fibroblast growth factor (BFGF), and platelet-derived growth factor B (PDGFB) mRNA in the PRP + ADSCs group from two to twelve weeks. In summary, the study suggests that combining ADSCs with PRP has a potent effect on improving skin grafts by promoting angiogenesis, increasing blood flow, and enhancing the quality of the transplanted skin tissue.

In a study by Meghan Samberg et al. [96], mesenchymal stem cells (ASCs) were isolated from human abdominal skin tissues. For PRP preparation, human blood units were obtained from 12 healthy donors. The research experiment involved creating full-thickness skin wounds with a diameter of 1.5 cm on the dorsum (back) of rats. Animals (N = 40) were distributed into five treatment groups: saline control group (250 mL of saline), PFP gel, PFP + ASCs gel, PRP gel, and PRP + ASCs gel treatments. This model was used to investigate the effects of co-administration of PRP stem cells on wound healing in an animal study. The findings of the study indicated an increase in the number of blood vessels in the wounds treated with PRP + ASCs gel compared to those treated with saline, showing that this combination treatment promoted angiogenesis. Additionally, PEGylated PRP hydrogels were found to serve as an excellent scaffold for supporting the growth and differentiation of ASCs over a 14-day period leading to improved wound healing outcomes.

A case report by Gian Marco Palini et al. [97] describes a customized surgical approach used to repair a grade IV abdominal incisional hernia. This approach involved the combined utilization of platelet-rich plasma (PRP) and mesenchymal stromal cells derived from the patient’s bone marrow. This patient-tailored surgical technique involving PRP and bone marrow-derived stromal cells applied to a biological mesh aims to optimize biocompatibility, reduce inflammation and adhesion formation, and promote wound healing. These efforts collectively contributed to an improved surgical outcome and may decrease the likelihood of hernia recurrence in the future.

The objective of a study conducted by Suk Ho Bhang et al. [98], was to assess whether the combined use of platelet-rich plasma (PRP) and human adipose-derived stem cells (hASCs) could enhance the effectiveness of hASCs in promoting skin wound healing. The study was conducted on athymic mice with skin wounds. These wounds were left untreated or treated with hASCs, PRP, or a combination of both. Firstly, human lipoaspirates were obtained from patients through elective liposuction. Additionally, whole blood was collected from healthy volunteers. Six-week-old female athymic mice were used for the study, and a full-thickness wound 1.8 × 1.8 cm^2^ was created on the back of each mouse. Rats were divided into four groups and wounds were treated with HCF (control), hASCs (1.8 × 10^6^ cells per defect), PRP (100 µL per defect), or hASCs (1.8 × 10^6^ cells per defect) suspended in 100 µL of PRP (hASC + PRP group). The study revealed that when human adipose-derived stem cells (hASCs) and platelet-rich plasma (PRP) were administered together, there was a significant increase in the formation of small blood vessels compared to the groups that did not receive any treatment, those treated with PRP alone, or those treated with hASCs alone. This finding suggests that the co-administration of PRP and hASCs has a beneficial effect on the healing of skin wounds by enhancing cell proliferation and angiogenesis.

The main findings from in vivo studies of surgical meshes combined with plasma-derived products and stem cells are summarized in Table 3. 

## 4. Concluding Remarks and Future Perspectives

The majority of studies examined in this review share some common drawbacks. These include a lack of in-depth discussion regarding long-term outcomes. Also, different protocols and methods for PRP preparation and stem cell isolation were used in these studies. In summary, given the high frequency with which hernia repair surgery is performed and despite the significant progress and evolution of surgical meshes, further efforts should be made in this direction to minimize recurrence and complication rates. Strengthening the process of neoangiogenesis and limiting the inflammatory reaction will enhance tissue regeneration and contribute to better integration of the mesh by the body. Further studies will help to evaluate the effect of PRP and its appropriateness as a scaffold for stem cell proliferation. Co-administration of PRP and stem cells has positive effects in wound healing and may eventually be acknowledged as a suitable treatment in regenerative medicine.

## Figures and Tables

**Figure 1 bioengineering-12-01182-f001:**
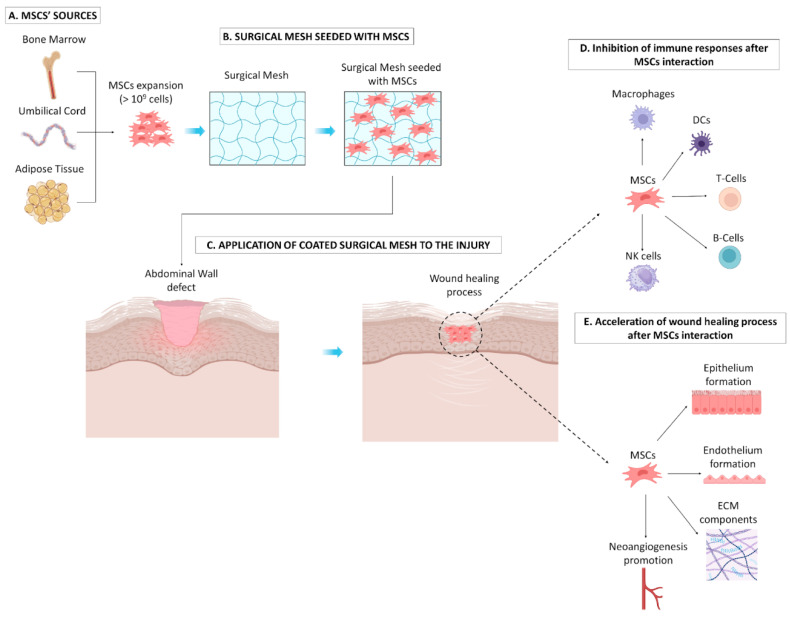
Effect of surgical mesh seeded with MSCs on the abdominal wall defect. (**A**) Initially, MSCs can be efficiently isolated from different sources (e.g., BM, UC, and AT) and can be expanded under in vitro culture conditions in great numbers. (**B**) Expanded MSCs can be seeded in surgical meshes to advance their properties, following implantation into the injury site (**C**). MSCs can promote acceleration of the wound healing process through the inhibition of topical inflammation exerted by immune cells, e.g., macrophages, DCs, T cells, B cells, and NK cells (**D**), while also reinforcing soft tissue reconstruction through the promotion of epithelium and endothelium formation, secretion of ECN components (e.g., collagen, sGAGs), and neoangiogenesis formation (**E**).

**Figure 2 bioengineering-12-01182-f002:**
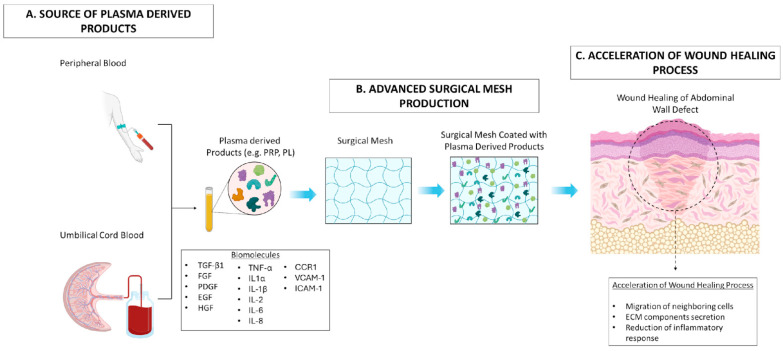
Effect of advanced surgical mesh implants coated with plasma-derived products in wound healing process. (**A**) Plasma-derived products including PRP and PL can be produced either from peripheral or umbilical cord blood with a defined process. (**B**) These products, rich in a great number of biomolecules, can be efficiently combined with the surgical mesh to advance their properties. (**C**) Finally, the produced surgical mesh can be used for the acceleration of the healing process in the abdominal wall defect through the migration of neighboring cells (e.g., SMCs, ECs, EpCs), the promotion of ECM component secretion (e.g., collagen, sGAGs, proteoglycans), and inhibition of the topical inflammatory response occurring at the injury site.

**Table 1 bioengineering-12-01182-t001:** In vivo studies of surgical meshes combined with stem cells.

Material Composition	Application	Outcome	Research Team
porcine acellular dermal matrix mesh + ASCs vs. mesh alone	36 Brown Norway rats ventral hernia repair follow up: 2 w	ASCs: ↑ neovascularization	Altman et al. (2010) [78]
Strattice^®^ mesh + ASCs vs. mesh alone	60 Norway rats ventral hernia defects follow up: 1, 2, 4 w	ASCs: ↑ neovascularization	Iyyanki et al. (2014) [79]
nanofibrous extracellular matrix (ECM) + MSCs vs. acellular mesh	15 rabbits inguinal hernia model follow up: 2 m	MSCs: ↑ vascularization ↓ adhesion formation	Zhang et al. (2017) [80]
macroporous meshes + hSVF vs. mesh + cell-free fibrin	28 male athymic rats full-thickness defect 1 × 1 cm follow up: 10, 21 d	SVF: ↓ vessel volume	O. Guillaume et al. (2020) [81]
monofilament polypropylene PP + MSCs vs. mesh alone	10 Large White pig congenital abdominal non-incarcerated hernias follow up: 1 m	MSCs: ↓ VEGF expression ↓ vascularization	Federica Marinaro et al. (2020) [82]
decellularized dermal scaffolds + BM-MSC vs. mesh alone	New Zealand white rabbits. abdominal wall defects follow up: 2 m	BM-MSC: ↑ tissue regeneration, ↑ cellular infiltration, ↑angiogenesis ↓adhesions ↓recurrence	Zhao et al. (2012) [83]
nanostructured membranes + MSCs vs. membranes	70 male Wistar R. norvegicus rats deep second-degree burns with a diameter of 2 cm follow up: 5, 30 d	MSCs: ↑ healing process ↑ angiogenesis ↑ epithelialization ↓ immune response,	Costa de Oliveira Souz et al. (2021) [84]
nanomesh + heMSCs vs. nanomesh alone	NSG mice pelvic organ prolapse follow up: 1, 6 w	eMSCs: ↓ inflammation ↑ enhance extracellular matrix (ECM) ↑ angiogenesis and ↑ expression of anti-inflammatory genes	Shayanti Mukherjee et al. (2020) [85]
Polyamide meshes + eMSC vs. mesh alone	74 CBH-rnu/Arc immunodeficient nude rats Subcutaneous pockets follow up: 7, 14, 30, 60, 90 d	eMSCs: ↑ tissue integration ↓ inflammatory responses	Ulrich et al. (2014) [86]
subcutaneously injected autologous ASCs vs. PBS (no scaffold)	27 Sprague Dawley 2 cm^2^ burn wound model follow up: 33 d	ASCs: ↑ VEGF expression ↑ CD31 expression ↑ angiogenesis	Xiaolong Zhou et al. (2019) [87]
fibrin + h BM-MSCs vs. fibrin vs. bovine serum albumin	rnu homozygous rats myocardial infarction follow up: 2 days, 5 w	BM-MSCs: ↑ capillary density ↑ VEGF	Naan F. Huang et al. (2009) [89]

d = days, w = weeks, m = months, ↑ increase, ↓ decrease.

**Table 2 bioengineering-12-01182-t002:** In vivo studies of surgical meshes combined with plasma-derived products.

Material Composition	Application	Outcome	Research Team
ProGrip mesh + autologous PRP	54 patients HH repair 10–20 cm^2^ follow up: 48 m	recurrence 3, 7%	Paranyak et al. (2021) [76]
polypropylene mesh vs. PP + PRP vs. PP + PRF	32 patients open hernia repair follow up: 1, 3, 6, 12 m	PRP/PRF: ↑ wound healing ↑ mesh integration rate	Popescu et al. (2021) [77]
mesh + PRP	12 patients paraesophageal hernia repair (>5 cm) follow up: 12–18 m	PRP: Good reflux control No complication No recurrence	James et al. (2023) [90]
pADM + PRP vs. pADM + saline	48 male Lewis rats 2 cm incisional defect-hernia repair model follow up: 2,4,6 w	PRP: ↑ microvessel formation ↑ expression of VEGF ↑ wound healing	Fernandez-Moure et al. (2021) [91]
pADM + PRP vs. pADM alone	28 male Lewis rat full-thickness abdominal wall defect → 28 days → hernia repair follow up: 3-m	PRP: ↑ neovascularization ↑ incorporation of the mesh ↓ inflammatory response	Jeffrey L. Van Ep et al. (2019) [16]
Strattice + PRP vs. Strattice + saline	42 male Lewis rats full-thickness abdominal wall defect → 28 days → hernia repair follow up: 2, 4, 6 w	PRP: ↑ neovascularization ↑ tissue deposition ↑ CD31 marker	Fernandez-Moure et al. (2021) [92]
Bio-AVR mesh + PRP vs. Bio-AVR	14 female pigs hiatal defect up 4 cm^2^ follow up: 7 m	PRP: ↓inflammatory response ↑ neovascularization ↑ collagen fibers	Cristian E. Boru et al. (2022) [93]
polypropylene mesh + PRP vs. mesh	hypoestrogenic rabbit models skin defect model follow up: 14, 28, 90 d	PRP: ↑ CD31 marker ↑ wound closure ↑ angiogenesis	Meutia AP et al. (2022) [94]

d = days, w = weeks, m = months, ↑ increase, ↓ decrease.

**Table 3 bioengineering-12-01182-t003:** In vivo studies of surgical meshes combined with plasma-derived products and stem cells.

Material Composition	Application	Outcome	Research Team
ADSCs + PRP gel formation 10^6^ cells/1 mL PRP gel were applied under the skin graft	72 male Lewis rats full-thickness wound defect 2.5 × 2.5 cm^2^ follow up: 2, 4, 12 weeks	ADSCs + PRP: ↑ CD31 positive cells ↑ VEGF + BFGF expression ↑ angiogenesis	Yiming Gao et al. (2020) [95]
hASC + PEGylated PRP hydrogels	40 male athymic rat full-thickness skin wounds 1, 5 cm diameter follow up: 8 d, 12 d	PRP + ASCs: ↑ angiogenesis	Meghan Samberg et al. (2019) [96]
autologous PRP BM-MSCs + biological porcine mesh	71 year-old female patient grade IV abdominal incisional hernia	PRP BM-MSCs: ↑ biocompatibility ↑ wound healing ↓ inflammation ↓ adhesion formation	Gian Marco Palini et al. (2017) [97]
PRP gel formation + hASCs 1.8 × 10^6^ cells/100 µL PRP per defect	female athymic mice full-thickness wound 1.8 × 1.8 cm^2^ defect follow up: 16 days	PRP + hASCs: ↑ cell proliferation ↑ angiogenesis	Suk Ho Bhang et al. (2013) [98]

d = days, w = weeks, m = months ↑ increase, ↓ decrease.

## Data Availability

No new data were created or analyzed in this study.

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
