# Peer review of "Advances in Innovative Surgical Implant Manufacturing for Hernia Repair and Soft Tissue Reconstruction"

_bioengineering, 2025, doi:10.3390/bioengineering12111182_

Round 1

Reviewer 1 Report

Comments and Suggestions for Authors

This review article provides valuable insights into the use of scaffold coatings with platelet-rich plasma (PRP), stem cells, or their combination in the context of hernia repair and soft tissue reconstruction. However, the review would benefit significantly from structural and visual improvements to enhance clarity and scientific impact. Here are some comments to improve the review article:

Current tables summarising experimental studies should be expanded to include: More detailed descriptions of scaffold materials, cell types used, model organisms, duration of study, and therapeutic outcomes. Move the bibliography citations to the end of each row. Separate tables for in vitro, in vivo, and combination (PRP + stem cells) studies would improve clarity.

Consider including a table that compares past and current technologies in scaffold-based regenerative therapies, advantages and limitations of PRP-only, stem cell-only, and combination approaches and status of each approach (preclinical, clinical trial phase, or clinical use).

Include a schematic figure, for example, by providing the mechanisms of action for PRP and stem cells. How are these integrated into surgical meshes. The expected biological outcomes (angiogenesis, reduced inflammation, better integration). A timeline or workflow of a typical preclinical/clinical application pipeline. This visual will help readers grasp the thematic framework of the review more intuitively.

Reframe certain sections to enhance flow and readability.

Ensure consistent terminology throughout (e.g., “MSC” vs. “MSCs”; “scaffold” vs. “mesh” where applicable).

Update the conclusion section to clearly highlight: Existing gaps in the literature, the future direction of combining PRP and stem cells in regenerative surgery. Emphasis on the need for standardized protocols and long-term outcome studies.

Ensure bibliographic consistency across all citations, especially in the tables.

Author Response

Dear Reviewer 1,

Initially, we would like to thank for the comprehensive peer-review performed for our submitted manuscript. Below you will find our responses to each of your valuable comments.

General Comment

This review article provides valuable insights into the use of scaffold coatings with platelet-rich plasma (PRP), stem cells, or their combination in the context of hernia repair and soft tissue reconstruction. However, the review would benefit significantly from structural and visual improvements to enhance clarity and scientific impact. Here are some comments to improve the review article:

Authors’ Response

Thank you for the peer-review performed to our manuscript. We have performed a number of revisisions, including adding figures, more explanatory tables, grammar-phrase revisions to the entire manuscript. In addition, manuscript scientific language was further improved by the mdpi-authors services-language editing. Hence, we believe by performing all these revisions the quality of manuscript has been significantly improved.

Reviewer’s Comments

Current tables summarising experimental studies should be expanded to include: More detailed descriptions of scaffold materials, cell types used, model organisms, duration of study, and therapeutic outcomes. Move the bibliography citations to the end of each row. Separate tables for in vitro, in vivo, and combination (PRP + stem cells) studies would improve clarity.

Authors’ Response

Based on the reviewer comment, we have performed the revision, regarding the tables improvement. We have added 5 tables, with the recommended structure to the manuscript, including the modern studies and applications of surgical implants coated either with PRP, stem cells or combination of those to advance their properties, for the hernia repair and soft tissue reconstruction.

Reviewer’s Comments

Consider including a table that compares past and current technologies in scaffold-based regenerative therapies, advantages and limitations of PRP-only, stem cell-only, and combination approaches and status of each approach (preclinical, clinical trial phase, or clinical use).

Authors’ Response

We have considered your comment, however we do think that to apply properly to this comment will extent significantly the time required for the revision performance. However, we have included tables in the main manuscript with the modern approaches of the surgical implants coated with prp, stem cells or combination of those.

Reviewer’s Comments

Include a schematic figure, for example, by providing the mechanisms of action for PRP and stem cells. How are these integrated into surgical meshes. The expected biological outcomes (angiogenesis, reduced inflammation, better integration). A timeline or workflow of a typical preclinical/clinical application pipeline. This visual will help readers grasp the thematic framework of the review more intuitively.

Authors’ Response

We appreciated much the reviewer’s comments, and for this purpose we have included two schematic figures. The first figure has been added at section 2.2 Stem cells, entitled “Figure 1. Effect of surgical mesh seeded with MSCs to the abdominal wall defect.”, explaining in this way the beneficial combination of stem cells and surgical implants to improve their properties and to advance the wound healing procedure for the hernia repair. The second figure has been introduced at section section 2.3 Plasma derived products, entitled “Figure 2. Effect of advanced surgical mesh implants coated with plasma derived products in wound healing process”, explaining the potential mechanism of how the plasma derived products can induce proper hernia repair, by reducing in parallel the host immune responses. We think, these 2 figures will improve further the quality of the manuscript, to be more informative for the majority of the readers.

Reviewer’s Comments

Reframe certain sections to enhance flow and readability.

Authors’ Response

Reframe of the sections has been performed to the entire manuscript, improving further quality of the manuscript. In addition, further scientific context and language polishing has been performed using the mdpi author’s services.

Reviewer’s Comments

Ensure consistent terminology throughout (e.g., “MSC” vs. “MSCs”; “scaffold” vs. “mesh” where applicable).

Authors’ Response

We have used consistent terminally throughout the entire manuscript. We thank the reviewer for the valuable comment.

Reviewer’s Comments

Update the conclusion section to clearly highlight: Existing gaps in the literature, the future direction of combining PRP and stem cells in regenerative surgery. Emphasis on the need for standardized protocols and long-term outcome studies.

Authors’ Response

We have updated the conclusion section to Concluding remarks and Future Perspectives, where we have included the future directions regarding the development of the surgical implants with advanced properties for hernia repair and soft tissue reconstruction.

Reviewer’s Comments

Ensure bibliographic consistency across all citations, especially in the tables.

Authors’ Response

We thank again the reviewer for the valuable comments. We have performed revision in the entire manuscript to ensure the bibliographic consistency across all sections of the manuscript.

Once again, we would like to thank the reviewer for the valuable comments. We have performed revisions to the entire manuscript and corrected any grammar and phrase errors. Also the scientific context of the review has been improved and the language has been further polished by mdpi-authors service – language editing. We think that the quality of the manuscript has been improved, hence can be processed to the next step of the publication process. If the reviewer feels that the manuscript requires any further revision, we are more than willing to perform, in order the manuscript to be processed to the next step of the publication process.

Yours sincerely,

Panagiotis Mallis

Reviewer 2 Report

Comments and Suggestions for Authors

The reviewed article presents an excellent summary of the most recent findings concerning bioengineered scaffolds coated with mesenchymal stem cells (MSCs) and platelet-rich plasma (PRP) that can be used for hernia repair, pelvic organ prolapse, wound healing and myocardial regeneration. It categorizes the articles by scaffold type, cell coating and application of the scaffolds and addresses the benefits of the MSCs and PRP on stimulating angiogenesis, minimizing inflammation and enhancing tissue integration. The authors reviewedarticles provided an adequate review of the studies however, it failed to give a good mechanistic understanding of the trials, their long-term outcomes or their ability to translate. They also failed to consider the variance in the protocols examples: the preparation and isolation methods of PRP and how the difference could limit direct comparisons between studies on the same topic. The conclusion noted these gaps, however, it would have been beneficial to offer standardized techniques of studies or sources of future research. To improve the article following comments should be address:

- The report summarizes results (e.g., increased angiogenesis, reduced inflammation) but does not address the possible biological mechanisms. 

- underscore the importance of using standardized protocols for PRP preparation (e.g., platelet concentration, activation mechanisms) and MSC isolation procedures (e.g., tissue source, culture conditions). This is required for reproducibility and translation to the clinical situation.

- many studies have short follow-up (e.g., weeks to months). Discuss other long-term risks, such as ectopic calcification, fibrosis or immunogenic response to allogeneic cells. Although not necessary, including data on functional recovery (e.g., biomechanical strength of repaired hernias) would be ideal. 

- Systematically tabulate or using meta-analysis the quantitative outcomes (e.g., vessel density, VEGF) across studies could help identify patterns (e.g., the efficacy of adipose-derived MSCs compared to bone marrow-derived MSCs).

- Discuss the new and exciting scaffold technologies (e.g., electrospun nanofibers, 3D-printed matrices) that may improve cell retention or controlled-release of bioactive factors.

- You should include schematic diagrams depicting the mechanisms (e.g., MSC paracrine effects) or flow diagrams of the PRP/MSC preparation methods to enhance clarity.

- Consistent terminology (e.g., "neovascularization" vs. "angiogenesis") must be assured.

- Update and incorporate landmark papers in the citations (e.g., 2023–2024 papers discussing PRP standardization).

Author Response

Dear Reviewer 2,

Initially, we would like to thank for the comprehensive peer-review performed for our submitted manuscript. Below you will find our responses to each of your valuable comments.

General Comment

The report summarizes results (e.g., increased angiogenesis, reduced inflammation) but does not address the possible biological mechanisms. 

Authors’ Response

Based on the reviewer comment, we have emphasized in this.

Reviewer’s Comments

underscore the importance of using standardized protocols for PRP preparation (e.g., platelet concentration, activation mechanisms) and MSC isolation procedures (e.g., tissue source, culture conditions). This is required for reproducibility and translation to the clinical situation.

Authors’ Response

Based on the reviewer comment, we have emphasized in this. Protocols for PRP preparation and MSC isolation must be standardized for future trials.

 Reviewer’s Comments

many studies have short follow-up (e.g., weeks to months). Discuss other long-term risks, such as ectopic calcification, fibrosis or immunogenic response to allogeneic cells. Although not necessary, including data on functional recovery (e.g., biomechanical strength of repaired hernias) would be ideal. 

Authors’ Response

We thank the reviewer for this valuable comment. Indeed, many studies have relatively short follow-up periods, which limits the ability to assess long-term outcomes.  Unfortunately, most of the currently available studies do not report extended follow-up. 

Reviewer’s Comments

Systematically tabulate or using meta-analysis the quantitative outcomes (e.g., vessel density, VEGF) across studies could help identify patterns (e.g., the efficacy of adipose-derived MSCs compared to bone marrow-derived MSCs).

Authors’ Response

We have considered your comment, however we do think that to apply properly to this comment will extent significantly the time required for the revision performance.  

Reviewer’s Comments

Discuss the new and exciting scaffold technologies (e.g., electrospun nanofibers, 3D-printed matrices) that may improve cell retention or controlled-release of bioactive factors.

Authors’ Response

Based on the reviewer comment, we have emphasized in this.

Reviewer’s Comments

You should include schematic diagrams depicting the mechanisms (e.g., MSC paracrine effects) or flow diagrams of the PRP/MSC preparation methods to enhance clarity .

 Authors’ Response

We have included 2 schematic figures in the text for this purpose.

 Reviewer’s Comments

Consistent terminology (e.g., "neovascularization" vs. "angiogenesis") must be assured

 Authors’ Response

We have used consistent terminally throughout the entire manuscript. We thank the reviewer for the valuable comment.

 Reviewer’s Comments

Update and incorporate landmark papers in the citations (e.g., 2023–2024 papers discussing PRP standardization).

Authors’ Response

Based on the reviewer comment, we have emphasized in this.

We think that with the performed revisions, the quality of the manuscript has been increased and can be further processed to the next step of the publication process.

We remain at your disposal if anything else is required by our side.

Yours sincerely,

Dr.Patsouris

Round 2

Reviewer 1 Report

Comments and Suggestions for Authors

Dear Authors,

You have successfully addressed all reviewer comments and significantly improved the manuscript’s structure, clarity, and visual presentation. The expanded tables, added schematic figure, and refined conclusion have greatly enhanced the scientific quality and readability of the review.

Reviewer 2 Report

Comments and Suggestions for Authors

the article has well revised and it is suitable for publication.